# NAD- and NADPH-Contributing Enzymes as Therapeutic Targets in Cancer: An Overview

**DOI:** 10.3390/biom10030358

**Published:** 2020-02-26

**Authors:** Alvinsyah Adhityo Pramono, Gulam M. Rather, Herry Herman, Keri Lestari, Joseph R. Bertino

**Affiliations:** 1Rutgers Cancer Institute of New Jersey, Rutgers, The State University of New Jersey, New Brunswick, NJ 08901, USA; alvinsyah.pramono@rutgers.edu (A.A.P.); gmr112@cinj.rutgers.edu (G.M.R.); 2Department of Pharmacology and Clinical Pharmacy, Faculty of Pharmacy, Universitas Padjadjaran, Sumedang 45363, Indonesia; lestarikd@unpad.ac.id; 3Center of Excellence in Higher Education for Pharmaceutical Care Innovation, Universitas Padjadjaran, Sumedang 45363, Indonesia; 4Division of Oncology, Department of Orthopaedic Surgery, Faculty of Medicine, Universitas Padjadjaran, Bandung 40161, Indonesia; herry_herman@yahoo.com; 5Department of Pharmacology and Medicine, Robert Wood Johnson Medical School, Rutgers, The State University of New Jersey, New Brunswick, NJ 08901, USA

**Keywords:** NAMPT inhibitor, NADK inhibitor, dihydrofolate reductase, IDH mutation, NAD/NADPH pool

## Abstract

Actively proliferating cancer cells require sufficient amount of NADH and NADPH for biogenesis and to protect cells from the detrimental effect of reactive oxygen species. As both normal and cancer cells share the same NAD biosynthetic and metabolic pathways, selectively lowering levels of NAD(H) and NADPH would be a promising strategy for cancer treatment. Targeting nicotinamide phosphoribosyltransferase (NAMPT), a rate limiting enzyme of the NAD salvage pathway, affects the NAD and NADPH pool. Similarly, lowering NADPH by mutant *isocitrate dehydrogenase 1/2* (*IDH1/2*) which produces D-2-hydroxyglutarate (D-2HG), an oncometabolite that downregulates nicotinate phosphoribosyltransferase (NAPRT) via hypermethylation on the promoter region, results in epigenetic regulation. NADPH is used to generate D-2HG, and is also needed to protect dihydrofolate reductase, the target for methotrexate, from degradation. NAD and NADPH pools in various cancer types are regulated by several metabolic enzymes, including methylenetetrahydrofolate dehydrogenase, serine hydroxymethyltransferase, and aldehyde dehydrogenase. Thus, targeting NAD and NADPH synthesis under special circumstances is a novel approach to treat some cancers. This article provides the rationale for targeting the key enzymes that maintain the NAD/NADPH pool, and reviews preclinical studies of targeting these enzymes in cancers.

## 1. Introduction

Sensitizing cancer cells to lower levels of NAD and NADPH pool without affecting normal cells is a novel anti-cancer strategy. The interdependence of different pathways of the pool is a difficult challenge to target them separately. Nicotinamide phosphoribosyltransferase (NAMPT), the rate-limiting enzyme of NAD salvage pathway, is the greatest contributor in generating NAD in mammals [1,2]. Overexpression of NAMPT has been implicated in several cancers [3,4,5]. On the other hand, nicotinate phosphoribosyltransferase (NAPRT) also produces NAD through the nicotinic acid pathway [1,6]. However, recent studies have shown that isocitrate dehydrogenase (IDH) mutations may result in downregulation of NAPRT [7,8,9].

NADPH is essential in controlling high levels of reactive oxygen species (ROS) in rapidly proliferating cancer cells [10] and in protecting dihydrofolate reductase (DHFR) from degradation [11]. Finally, NADPH is used to generate the oncometabolite D-2-hydroxyketoglutarate (D-2HG) [12,13]. These findings have stimulated efforts to target these pathways in cancer cells. In this review, we describe the approaches to lowering NAD and NADPH levels or both in many types of cancer that may allow for selective toxicity.

## 2. Targeting NAD Synthesis

Cancer cells produce energy through aerobic glycolysis rather than conventionally developing energy through the citric acid cycle and respiratory chain, a phenomenon known as the Warburg effect [14]. Aerobic glycolysis requires NAD as electron recipient during glyceraldehyde-3-phosphate dehydrogenase (GAPDH) catalysis of glyceraldehyde-3-phospate into diphosphoglycerate, and generates inorganic phosphate for ATP production. Gluconeogenesis utilization also requires NAD which is converted to NADPH during the cytosolic conversation of malate intermediate into oxaloacetate. NAD depletion is seen as disaster for cancer cells, as both glucose production and glucose utilization will come to a halt. The amount of energy produced from aerobic glycolysis is less than the energy produced by the normal respiratory chain. However, cancer cells are able to produce significant amount of energy and biomass to support cancer survival in a short period of time [6,14,15].

When cancer cells become quiescent, the citric acid cycle and electron transport chain are preferred compared to aerobic glycolysis [16]. Excessive amount of NADH obtained from glycolysis is utilized to convert dihydroxyacetone phosphate (DHAP) into glycerol-3-phosphate (G3P). G3P donates electrons for the oxidative phosphorylation to maintain mitochondrial integrity and ATP production in quiescent cancer cells [16]. Recent evidence showed that oxidative phosphorylation is upregulated in some cancers, including breast cancer, Hodgkin lymphoma and RB1-deficient cancers [17,18,19,20,21]. These results indicated that targeting NADH pool could affect oxidative phosphorylation in these cancer types [16,22].

NAD levels in cancer cells are generated through three separate pathways: the nicotinic acid (NA) pathway, the de novo pathway and the NAD salvage pathway (Figure 1) [6,23]. The precursors for NA and the de novo pathway are nicotinic acid and dietary tryptophan, respectively [24,25]. NAPRT converts NA into nicotinic acid mononucleotide (NAMN). Tryptophan, is converted to quinolinic acid, and is incorporated into the NA pathway as NAMN to generate NAD. Sirtuins, poly (ADP-ribose) polymerases (PARPs) and CD157, NAD consumers, recycle NAD by providing nicotinamide (Nam) as the precursor of the NAD salvage pathway. The NAD salvage pathway is the major source of NAD in humans that depends on NAMPT, as it is the rate-limiting enzyme [26].

Despite conventional pathways that have been explored, Charles Brenner found a new mechanism of generating NAD using synthetic precursors following the emergence of vitamin B3 supplementation to prevent pellagra in 2004 [27]. Both nicotinamide riboside (NR) and nicotinic acid riboside (NAR) are exogenous sources of NAD production [28]. NR is incorporated to the NAD salvage pathway after conversion to nicotinamide mononucleotide (NMN) [28,29]. NAR becomes a part of the NA pathway after it is converted to NAMN [28,29]. Both reactions are catalyzed by nicotinamide riboside kinase 1/2 (NRK1/2) [27,28,29] (Figure 1). NMN supplementation is also available, however in order to be internalized in the cell, it has been postulated that NMN must be converted to NR by cell membrane protein CD73 (also known as ecto-5-nucleotidase) [28,30].

Early on, elevated NAD levels exert a protective effect against oncogenesis [31,32,33,34], later on, elevated NAD levels enhance cancer cell survival and progression [35,36,37]. Demonstrating the effect of restricting the NAD pool in halting cancer progression requires the inhibition of vital pathways, and the reduction of redundancy [38]. This may be achieved in cancer cells that are deficient in one of the NAD biogenesis pathways.

Some cancers, such as glioblastoma, chondrosarcoma, leukemia and colorectal cancer, have mutations in *isocitrate dehydrogenase 1/2 (IDH1/2)* [13,39,40,41]. Mutant IDH1/2 uses NADPH to convert α-ketoglutarate (α-KG) into D-2HG, an oncometabolite that causes hypermethylation at CpG islands or loss of exon 1 expression in NAPRT [7,8,9,39,40]. As a consequence, inhibition of NAPRT enzyme activity forces these cells to mainly depend on the NAD salvage pathway to generate NAD (Figure 1) [7]. Furthermore, NAMPT is frequently amplified in some cancers which may also affect the NAD pool [3,7,42,43]. Conversely, NAMPT-specific inhibitors significantly deplete NAD levels and subsequently suppress cancer cell proliferation [7,44,45,46].

### 2.1. Therapeutic Role of NAMPT in Cancer

NAMPT, a dimeric class of type II phosphoribosyltransferases, catalyzes NMN from Nam and 5’-phosphoribosyl-1-pyrophosphate (PRPP) [1]. NAMPT can be found in both intracellular and extracellular environments, including cytoplasm, blood, cerebrospinal fluids, adipose tissue, hepatic tissue, pancreatic tissue and almost every organ in the human body [1,47]. Phosphorylation at His247 increases the affinity of NAMPT to Nam and NAMPT enzymatic activity for more than 1000 folds [1,48,49].

Increased serum concentrations of NAMPT have been linked with diseases such as obesity, non-alcoholic fatty liver disease, diabetes mellitus and in particular, cancers [35,36]. Colorectal cancer, ovarian cancer, breast cancer, prostate cancer, gastric cancer, melanoma and myeloma were found to overexpress NAMPT [15,44,50,51,52,53]. An increased NAD pool, as the consequence of NAMPT overexpression, was also associated with chemotherapeutic resistance [50,54]. Therefore, targeting NAMPT in tumors lacking NAPRT has been identified as an anti-cancer drug target. NAMPT knock-down has successfully sensitized cancer cells to increased ROS and cell death [44].

In vitro studies have shown promising results using a NAMPT inhibitor in cancer cells, especially the *IDH1/2* mutant cancer cell lines (Table 1). MGG119, MGG152, BT142 primary glioblastoma cell lines; HT1080, 30T and SW1353 chondrosarcoma cell lines; SNU484, SNU668, SNU1750, MKN1 and Hs746T gastric cancer cell lines which have mutations in IDH1 [7,8,9], were sensitive to NAMPT inhibition. NAMPT inhibitors not only have shown promising effect as single-agent therapy, but were also found to sensitize other modalities of cancer treatment in both in vitro and in vivo experiments [45,55,56,57], as shown in Table 2 and Table 3.

As preclinical studies with NAMPT showed encouraging results [73,74,75], the clinical efficacy and safety of NAMPT inhibitors has been tested in cancer patients. There are 8 registered trials for NAMPT inhibitors in cancers in clinicaltrials.gov. The first registered NAMPT inhibitor trial in humans (NCT00003979) was initiated in 1999 and withdrawn in 2012. Some of the earliest trials of NAMPT inhibitor were phase I studies utilizing the NAMPT inhibitor CHS828, to treat solid tumors. They established the recommended dose for CHS828; however, studies were not recommended for further trials due to disease progression and severe adverse events experienced by study subjects [76,77,78]. A phase II clinical trial utilizing the NAMPT inhibitor, FK866 for cutaneous T-cell lymphoma was terminated due to failure to achieve remissions and serious adverse events [79]. Disappointing results were attributed to cellular uptake of extracellular NAD precursors such as vitamin B3, nicotinic acid riboside and dietary tryptophan [80]. In these studies patients were not selected for IDH mutations. Importantly, D-2HG accumulation and NAPRT hypermethylation that occurs in mutant *IDH1/2* cancers could be potential candidates for NAMPT inhibitor trials as they lack NA pathway (Figure 1).

### 2.2. Mutant Isocitrate Dehydrogenases Inhibit NAD Production in Cancer

Isocitrate dehydrogenase (IDH) converts isocitrate into α-ketoglutarate (α-KG) in the TCA cycle with NADP^+^ as its cofactor. IDH exists in three isozymes, namely IDH1, IDH2 and IDH3. IDH1, located in the cytosol and peroxisome, is encoded by the *IDH1* gene on chromosome 2q34, while IDH2 is localized in the mitochondria and is encoded by the gene located on chromosome 15q26.1 [41]. Both enzymes are NADP-dependent, share high degree of homology and conduct a reversible reaction to maintain the isocitrate pool [12]. Unlike the other two isozymes, IDH3, localized in the mitochondria, exists as a heterocomplex enzyme consisting of α, β, and γ subunits that are encoded by *IDH3A* (15q25.1), *IDH3B* (20p13), and *IDH3C* (Xq28) genes, respectively [41]. IDH3 is activated by adenosine diphosphate (ADP) and inhibited by adenosine triphosphate (ATP). In order to ensure the continuity of TCA cycle, IDH3 conducts the irreversible conversion of isocitrate to maintain the α-KG pool [12,41,81].

IDH mutations have been associated with several cancers, in particular, WHO grade II/III and secondary glioblastoma multiforme (GBM), acute myeloid leukemia (AML), intrahepatic cholangiocarcinoma, gastric cancer and cartilaginous tumors [9,82,83,84,85]. Most of mutant *IDH1* tumors have a single, missense mutation at codon 132 that changes arginine to histidine (IDH1^R132H^). The other base substitution with arginine and other amino acid residue at the same position has also been reported [12,13,81]. Similarly, the most frequent hot spots for mutant *IDH2* tumors are arginine at other positions, R172K and R140Q. The accumulation of NADPH derived from isocitrate conversion to α-KG is used to maintain the pool of reduced glutathione (GSH), peroxiredoxin and catalase tetramers that scavenge ROS [12,41]. NADPH may benefit mutant IDH cancers as it assists mutant IDH in generating D-2HG from α-KG to promote oncogenesis [37,50].

Mutant *IDH1/2* utilizes NADPH to convert α-KG to D-2HG and NADP. Physiologically, D-2HG is produced in a very small amounts as a result of errors in metabolism [12]. Thus, the mutant *IDH1/2* displays a new ability to convert α-KG to D-2HG. D-2HG competitively binds dioxygenases, including the ten-eleven translocation (TET) family of 5-methylcytosine hydroxylases, the Jumonji-domain containing histone-lysine demethylases (JMJ-KDMs), the AlkB family of dioxygenases, the hypoxia-inducible factor (HIF) prolyl 4-hydroxylases and asparaginyl hydroxylase, and the collagen prolyl and lysine hydroxylases. D-2HG modulates epigenetic regulation and increases the chance of tumorigenesis in mutant IDH1/2 cells [12,41].

D-2HG inhibits the α-KG dependent dioxygenases, TET2 and JMJD-KDMs causing hypermethylation in the mutant IDH1/2 tumors [81]. One of the causes of tumorigenesis is chromatin modification in mutant IDH tumors. TET2, a tumor suppressor gene as well as a part of the TET family, is usually mutated and causes hypermethylation in AML cancers. Hypermethylation was observed at CCCTC-binding factor (CTCF)-binding sites, insulator sequences that block the interaction between enhancers and promoters, causing chromatin modifications to modulate the gene expression in the tumorigenesis process [12,13,41,81].

## 3. Targeting NADPH Formation

### 3.1. Therapeutic Role of NAD Kinase

Cancer cells depend on the availability of NADPH to provide defense against reactive oxygen species (ROS) that is generated from cancer proliferating activity [86]. The formation of NADPH is catalyzed by NAD Kinase (NADK) and dehydrogenases which convert NAD to NADP and NADP to NADPH, respectively [87]. NADK was discovered by Kornberg in 1950 after Vestin, von Euler and Adler observed NADP formation in yeast [87]. Since then, NADK has been found in algae, high-level plants, and animals, including monkey, rabbit, chicken liver and ground squirrels [87,88,89]. NADK was found in almost all human organs except skeletal muscle. It is localized in the cytoplasm and recently a mitochondrial enzyme has been described [90,91,92,93]. The cytoplasmic enzyme consists of four subunit polypeptides and optimally works at temperature of 55 °C and pH 7.5 [90,91]. NADK requires NAD and ATP and a divalent metal ion such as zinc, manganese and magnesium [90]. NADK has also been observed to play a role in physiology and pathophysiology of diseases, including cancers [92,94,95,96]. NADK generates NADPH that is important to scavenge ROS, and promote cell longevity [97,98]. Thus, targeting NADK in cancer has been recognized as another potential anti-cancer target [10].

NADK is inhibited by thionicotinamide adenine dinucleotide (NADS) and thionicotinamide adenine dinucleotide phosphate (NADPS) [10,11,86]. The prodrug thionicotinamide, (TN) is converted into NADS and NADPS intracellularly and acts as dual inhibitor of NADK and glucose-6-phosphate dehydrogenase (G6PD) that catalyzes NADPH production. Reducing intracellular NADPH results in an increase in ROS [10,11]. Reduction of NADP^+^ by dehydrogenases produces NADPH the substrate for glutathione reductase and mediates the production 5-phosphoribosyl-1-pyrophosphate (PRPP), the substrate for the synthesis of purine and pyrimidine. Depletion of NADPH puts cancer cells in higher oxidative state and limits availability of nucleic acid for DNA replication.

TN successfully delayed tumor growth in xenograft mouse models of colon cancer and diffuse large B cell lymphoma (DLBCL) and in vitro synergized with other ROS-inducing chemotherapeutic agents [10]. As mutant IDH requires NADPH to produce D-2HG, thus, a NADK inhibitor might be a novel therapeutic approach for treating mutant IDH cancer cells to decrease the NADPH pool which may not be sufficient to facilitate the production of the D-2HG oncometabolite.

### 3.2. NADPH Stabilizes Dihydrofolate Reductase (DHFR)

DHFR catalyzes the reduction of folic acid and dihydrofolate (H_2_F) to tetrahydrofolate (H_4_F), utilizing NADPH as a cofactor and is important enzyme for de-novo- purine and thymidine biosynthesis [99]. The discovery of methotrexate as a competitive inhibitor of DHFR opened many avenues for cancer therapeutic purposes [100]. To understand the mechanism of methotrexate resistance in cancers or up-regulation of DHFR in such cancers, it became evident that the interaction of DHFR with its co-factor NADPH plays a key role to regulate its interaction as well as its expression [11,101].

DHFR exists in two interconvertible conformers, one binding to its mRNA and regulates its translation and other conformer binds to NADPH to regulate its function as well [99]. The binding of methotrexate to DHFR-NADPH conformer leads to inhibition of its catalytic function, however binding to other conformer was found to shift the chemical equilibrium towards releasing the DHFR from its own mRNA and thus resulting in DHFR translation. Different studies [101,102] supported that the NADPH binding domain regulates DHFR translation. DHFR mutants (both catalytic or NADPH binding) alter its binding with its substrate or co-factors and modify its regulation [99,101]. Recently our group showed that targeting NADK by TN caused a decrease of NADPH which destabilized DHFR and caused its degradation [10,11]. Thus, lowering levels of DHFR may play a role in the anticancer activity of TN [10,11,99,103]. Targeting the NADPH pool via DHFR inhibition opens another window in cancer therapeutics.

### 3.3. Methylenetetrahydrofolate Dehydrogenase (MTHFD) Maintains NADP/NADPH Pools in Cancer

MTHFD2 is an enzyme that is required for tumor cell survival [104]. MTHFD2 is highly expressed in the mitochondria and nucleus of multiple types of tumor cells, but not in non-transformed cells [104]. The only non-transformed cell induction of MTHFD2 was observed in lymphocyte activation, indicating the role of MTHFD2 in normal hematopoietic cells as well as in the tumorigenesis process of hematologic malignancies [105].

MTHFD2 plays a significant role in the mitochondrial folate-dependent one carbon (1C) metabolism by oxidizing the 1C unit and recycling folate cofactor needed by the enzyme serine hydroxymethyltransferase 2 (SHMT2) [104,105]. Both reactions are highly important to ensure rapid cancer proliferation [104,105]. MTHFD2 exists as two isozymes, MTHFD2 and MTHFD2L. Both isozymes are bifunctional enzymes that are responsible for the NAD-dependent CH_2_-THF dehydrogenase and CH^+^-THF cyclohydrolase reactions within the mitochondria. The cytosolic counterpart, MTHFD1 is a trifunctional enzyme having dehydrogenase, cyclohydrolase and formyltetrahydrofolate synthetase activities. Recently another isozyme MTHFD1L, another mitochondrial enzyme was stuided in colorectal and bladder cancers, is a monofunctional enzyme that catalyses the formation of formate and THF cofactor regeneration. MTHFD1L enhanced proliferation in some cancer tissues, as knock down of MTHFD1L slowed the tumor growth [106,107].

The reactions catalysed by MTHFD2 and its isozyme are NAD-dependent [108]. MTHFD2 is critical in maintaining NADP/NADPH production as it showed specific redox cofactor activity with NAD and NADP [108,109]. Suppression of MTHFD2 disturbed NADPH pools and enhanced apoptosis due to accumulated oxidative stress [110]. Therefore, NAD depletion might also interfere with folate metabolism and cause cancer cell death.

### 3.4. Serine Hydroxymethyltransferase (SHMT) Maintains NADP/NADPH Pools in Cancer

SHMT2 is a mitochondrial enzyme that converts serine into glycine simultaneously converting tetrahydrofolate to 5,10-methylene-H_4_F [105,111]. SHMT2 plays an important role in folate metabolism as serine is a dominant source of 1C unit [111]. Increased regulation of mitochondrial 1C elevates the production of 1C unit, biosynthesis of adenosine, guanosine, thymidylate and remethylation of homocystein to support methionine cycle [111,112]. The accumulated 1C units are important to support folate metabolism which regenerates the redox cofactors NADPH, NADH and ATP [112].

Overexpression of SHMT2 has been found in cancers and associated with poor prognosis in cancer patients [111,113]. However, it is not sufficient to cause malignant transformation alone [114]. Increased expression in mRNA and protein of SHMT2 was found in hepatocellular carcinoma cell lines. Downregulating SHMT2, either by shRNA knock down, SHMT2 silencing, or inhibition using L-mimosine, suppressed tumor growth and increased chemotherapeutic sensitivity [114,115]. SHMT2 knockdown leads to decreased NADPH and impaired cancer survival [116]. Impairment of folate metabolism will decrease the production of NADPH, NADH, ATP and further cause cell death [112]. Therefore, SHMT2 is a potential target in cancer therapeutics. Targeting SHMT2 disturbs the homeostasis of NADP/NADPH pool which is important in cancer survival.

### 3.5. Targeting Aldehyde Dehydrogenase (ALDH) Sensitizes Cancer

Aldehyde dehydrogenases are groups of NAD-dependent enzymes in the ALDH superfamily that consists of 19 different NADP-dependent ALDH enzymes. ALDHs are localized in both cytosol and mitochondria [117]. Crystallographic analysis has shown that ALDH has a specific coenzyme NAD binding domain. This implies that NAD is a significant component required by ALDH to execute its functions in retinoic acid biosynthesis, GABA neurotransmitter metabolism and alcohol metabolism [117].

ALDH1A3, a member of ALDH1A subfamily, is associated with cancer incidence, progression, prognosis and chemotherapeutic resistance [117,118,119]. The gene encoding ALDH1A3 consists of 16 exons and is located on chromosome 15q26.3. It is localized in the cytoplasm, nucleus and mitochondria [119,120]. This enzyme converts all-trans-retinal into retinoic acid [119,121,122]. ALDH catalytic reactions produce vast amount of cytosolic NADH as the source of ATP generation in cancers [123]. ALDH1A3 also plays a role in generating acetic acid from acetaldehyde in glycolysis and gluconeogenesis as well as metabolizing amino acids, exogenous chemical substances via cytochrome P450 and lipid peroxidase product [119,121]. ALDH1A3 catalyzes peroxidation, a process that produces fatty aldehyde, an intermediate compound in producing fatty acid and NADPH [119,124]. ALDH1A3 indirectly regulates apoptosis through retinoic acid pathway by increasing expression of caspase-7 and caspase-9 [119,125].

ALDH1A3 is also expressed in some cancers, including pancreatic cancer, ovarian cancer and high-grade gliomas [119,126,127,128]. Hypermethylation of ALDH1A3 promoter region was found to be one of the underlying mechanisms in some cancers [129,130,131,132]. Overexpressing ALDH1A3 is associated with cancer stem cell characteristics by increasing cancer cell proliferation. ALDH1A3 knockdown decreased tumorigenesis [119,133,134,135]. An increased level of ALDH1A3 also correlates with chemotherapeutic resistance [119]. Therefore, targeting ALDH1A3 in cancer is of great interest.

Inhibition of ALDH cause ATP depletion that inhibits tumor growth in both in vitro and xenograft models [124,132]. Knockdown of ALDH significantly decreased NADH and ATP production in cancer [124,136]. Downregulation of ALDH1A3 sensitized cancer cells, prolonged G1 phase, shortened S phase and decreased cancer cell survival both in vitro and in vivo [135,137]. ALDH1A3 is a prominent target for chemotherapy in cancer as its inhibition disrupts NADH and ATP pools, shuts the power factory off in cancer, and sensitizes cancer cell death.

## 4. Conclusions

Exploring NAD biogenesis for cancer therapeutics had been studied for decades [24,25,35,91,92,94]. The NAD pool in cells is contributed by the simultaneous functioning of various pathways as mentioned in this review, viz-a-viz, NA pathway, de novo pathway and NAD salvage pathway. The malfunctioning of the enzymes in any of these pathways may alter the NAD pool of cells. Cancer cells attempt to maintain the NAD pool. In some cancers, the rate limiting enzyme of one of the NAD biogenesis pathways may be over expressed or a mutation in one pathway may alter the turn-over of the other pathway to sustain the NAD pool. Different metabolic enzymes including NAMPT, NAPRT, IDH, NADK, DHFR, MTHFD, SHMT and ALDH, are dysregulated in various cancer types, making them potential anti-cancer targets. Mutations in some NAD and NADPH metabolic enzymes cause epigenetic regulation and affect expression of different genes involved in different cellular processes. Thus, targeting different rate limiting enzymes alone or in combination opens many therapeutic opportunities to treat cancers.

## Figures and Tables

**Figure 1 biomolecules-10-00358-f001:**
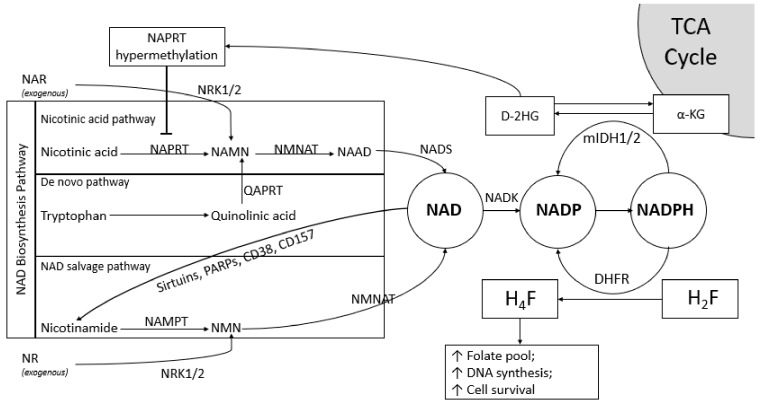
The association between *IDH1/2* mutation and NAD Production in Cancer. Mutant IDH1/2 converts α-ketoglutarate (α-KG) into D-2 hydroxyglutarate (D-2HG), an oncometabolite that causes hypermethylation at CpG islands of NAPRT promoter region, results in nicotinic acid pathway inhibition. Cancer cells maintain the adequate level of NAD pool through NAD salvage pathway and NADPH generation. NADPH is used as a substrate to generate D-2HG to promote oncogenesis. One of the possible mechanisms for cancer cell survival is achieved through catalytic activity of dihydrofolate reductase (DHFR) that maintains folate pool and induces DNA synthesis. TCA: tricarboxylic acid; NAPRT: nicotinate phosphoribosyltransferase; NAMN: nicotinic acid mononucleotide; NAAD: nicotinic acid adenine dinucleotide; NADS: NAD synthase; QAPRT: quinolinate phosphoribosyltransferase; NMN: nicotinamide mononucleotide; NAR: nicotinic acid riboside; NR: nicotinamide riboside; NRK1/2: nicotinamide riboside kinase 1/2; H_2_F: dihydrofolate; H_4_F: tetrahydrofolate.

**Table 1 biomolecules-10-00358-t001:** Performance of NAMPT inhibitors in NAPRT-deficient/depleted cancer cell lines.

	Cell Lines	IDH Mutation	NAPRT Expression	Sensitivity to NAMPT Inhibitors	NAD	Refs.
Glioblastoma	MGG119	IDH1^R132H^	Depleted	Highly sensitive	Decreased	[8,58]
	MGG152	IDH1^R132H^	Depleted	Highly sensitive	Decreased	
	BT142	IDH1^R132H^	ND	Highly sensitive	Decreased	
Gastric cancer	Hs746T	N/A	Depleted	Highly sensitive	Decreased	[4]
	SNU1750		Depleted	Highly sensitive	ND	
	MKN1		Depleted	Highly sensitive	ND	
	SK4		Normal	Resistant	ND	
	SNU484		Depleted	Highly sensitive	ND	
	SNU668		Depleted	Highly sensitive	Decreased	
	YCC11		Decreased	Moderately sensitive	ND	
	GA077		Depleted	Highly sensitive	Decreased	
Fibrosarcoma	HT1080	IDH1^R132C^	Depleted	Highly sensitive	Decreased	[8,9,58,59]
Melanoma	30T	IDH1^R132C^	ND	Highly sensitive	ND	[8]
Chondrosarcoma	SW1353	IDH2^R122S^	Depleted	Highly sensitive	ND	[8,9]
	JJ012	IDH1R^132G^	Depleted	Highly sensitive		
	L835	IDH1^R132C^	Increased	Moderately sensitive		
	L2975	IDH2^R132W^	Depleted	Highly sensitive		
	CH2879	IDH1/2 wt	Increased	Highly sensitive		
	NDCS-1	IDH1/2 wt	Depleted	Highly sensitive		
	CH3573	IDH1/2 wt	Decreased	Highly sensitive		
	L325b	IDH1/2 wt	Increased	Resistant		
	MCS170	IDH1/2 wt	Decreased	Resistant		
	OUMS27	IDH1/2 wt	Decreased	Moderately sensitive		
Non-small cell lung cancer	H460	ND	Depleted	Highly sensitive	ND	[59,60]
	A549	ND	Decreased	Highly sensitive		
Multiple myeloma	H929	ND	Depleted	Highly sensitive		
Glioblastoma	U251MG	ND	Depleted	Highly sensitive		
Pancreatic cancer	Mia-PaCa2	ND	Depleted	Highly sensitive		

N/A: not available; ND: not determined.

**Table 2 biomolecules-10-00358-t002:** Performance of NAMPT inhibitor in combination in vitro.

Cancer/Cell Lines	Drug Combination	Study Result	Refs.
*Fibrosarcoma* HT1080	2.5 nM FK-866/GMX-1778 + 200 μM Temozolomide	Decreased cell viability Decreased NAD Increased apoptosis Increased ROS	[58,61]
*Glioblastoma* MGG119, MGG152, BT142, U251	5/10 nM FK-866/CHS-828 + 100 μM Temozolomide		
*Ewing Sarcoma* TC32, TC71	5 nM Daporinad (FK-866) + 5 nM Niraparib	Decreased cell viability Decreased NAD	[62]
*Pancreatic ductal adenocarcinoma* PaTu8988t, Panc-1	6.25 nM STF-118804 + 5 nM Gemcitabine	Decreased cell viability	[45]
	6.25 nM STF-118804 + 5 nM Paclitaxel		
	6.25 nM STF-118804 + 300 nM Etoposide		
*Gastric cancer* BCG823	1 nM FK-866 + 3 μg/mL 5-Fluorouracil	Decreased cell viability Increased apoptosis	[3]
*Non-small cell lung cancer* A549, H1299	150 nM Pemetrexed for 48 h + 8 nM GMX-1777 for the next 24 h	Decreased cell viability Decreased ATP Decreased NAD Increased PARP expression	[60]
*Triple-negative breast cancer* CAL51	10/100 nM FK866 + 0.1/1/10 µM Olaparib	Decreased cell viability Decreased NAD Increased apoptosis	[63]
*Leukemia* MV4-11	100 nM KPT-9274 + 10 nM Venetoclax	Decreased cell viability Decreased NAD	[64]
Jurkat	10 nM FK-866 + 5/10 μM Etoposide	Decreased cell viability Decreased PARP-1 expression Increased apoptosis	[65]
MOLT-4			
OCI/AML3, Mec.1	3 nM APO866 for 48 h + 10 μM verapamil/1 μM Cyclosporin A/10 μM PGP-4008 for the next 48 h	Decreased cell viability Increased apoptosis	[66]
LAMA-84, RPMI-8226	3 nM APO866 for 48 h + 1 μM Cyclosporin A for the next 48 h		
*Leukemia* Jurkat, PEER, H9, Namalwa	0.1/3.2/4 nM APO866 + 0.1/6.3/100 ng/mL TRAIL	Decreased cell viability Decreased ATP Decreased NAD Increased cell death Increased autophagy	[67]
*Multiple myeloma* RPMI-8226/S, U266, MM1S, MM1R, ANBL6	1 nM/3 nM FK-866 + 5 nM/10 nM Bortezomib	Decreased cell viability Decreased NAD Increased apoptosis	[68]
*Waldenström macroglobulinemia* BCMW.1, MWCL.1	3 nM FK-866 + 1 μM Ibrutinib	Decreased cell viability Decreased ATP Decreased NAD Increased apoptosis	[64,69]

**Table 3 biomolecules-10-00358-t003:** Performance of NAMPT inhibitor in combination in vivo.

Cancer/Cell Lines	Drug Combinations	Study Result	Refs.
*Non-small cell lung cancer* A549, H1299	150 mg/kg/day GMX-1777 for 5 days, i.m. + 600 mg/kg/day Pemetrexed for 5 days, i.p.	Decreased tumor volume Increased body weight	[60]
*Ewing Sarcoma* TC32, TC71	25 mg/kg GNE-618 p.o. + 50 mg/kg Niraparib p.o.	Decreased tumor volume Increased survival	[62]
*Neuroendocrine tumor* GOT1	100 mg/kg GMX-1778, three weekly doses, p.o. + 7.5 MBq ^177^Lu-DOTATATE, single dose, i.v.	Decreased tumor volume Increased survival	[70]
*Prostate cancer* PC3	4 injections 10 mg/kg of APO-866 for 2 days + 4 fractions 3 Gy radiation for 4 days	Delayed tumor growth Increased survival	[71]
*Triple-negative breast cancer* CAL51	6 mg/kg FK866 + 15 mg/kg Olaparib for 5 days, followed by 2 days of no treatment, maintained upon study completion	Decreased tumor volume Increased body weight	[63]
*Ovarian cancer* OVCAR-3	10 mg/kg/day FK866, i.p. + 15 mg/kg APCP, i.p., every other two days for 28 days	Reduced NAD, NMN, ATP Increased cancer necrotic area (Ki67^+^ staining) Increased survival	[72]
*Waldenström macroglobulinemia* BCMW.1	30 mg/kg/day FK866 for 4 days in a week (with 3 days off), repeated for 3 weeks + 0.5 mg/kg/day Ibrutinib for 5 days, repeated for 3 weeks	Decreased tumor volume Increased survival	[69]
	100 mg/kg/day KPT-9274 for 5 consecutive days/week for 3 weeks + 25 mg/kg Bendamustine, one single dose/week for 2 consecutive weeks	Highly sensitive Increased apoptosis Decreased tumor volume	[64]
*Multiple myeloma* MM1S	30 mg/kg FK-866 for 4 days repeated weekly for 3 weeks, i.p. + 0.5 mg/kg Bortezomib biweekly for 3 weeks, s.c.	Decreased tumor volume Increased survival	[68]

i.m.: intramuscular; i.p.: intraperitoneal; p.o.: peroral; i.v.: intravenous; s.c.: subcutaneous.

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
