# Peer review of "NAD- and NADPH-Contributing Enzymes as Therapeutic Targets in Cancer: An Overview"

_biomolecules, 2020, doi:10.3390/biom10030358_

Round 1

Reviewer 1 Report

The authors have a prepared a concise and maybe too brief review discussing what has been reported on the therapeutic potential of NAD/NADP biosynthesis enzymes in cancer. Overall, this is a nicely written review but is considered maybe too brief as some aspects are neglected.

Concerns minor to be considered:

1) Sections 2.1, 3.1 and 3.3 are all titled “Therapeutic role of NAMPT in Cancer”. This must be an error.

2) Section 2 has only one sub-section. Might be worthwhile to expand.

3) There is some effort link NAD changes to the impact on NAD-requiring enzymes – some of this idea is shown in Table 2. However, much is left out. Other things to consider include the impact on ALDH enzymes, known to be important in cancer stem cells (such as ALDH1A3), the impact in DNA repair, the impact on epigenetics, the impact on mitochondrial function? And, how do these effects impact clinical attempts at combination therapy, as in Tables 2 and 3?

Reviewer 2 Report

The manuscript “NAD and NADPH contributing enzymes as therapeutic targets in cancer: an overview” by Pramono et al. reviews the roles of NAD/NADPH metabolyzing enzymes in tumors.

The first suggestions is to change significantly the abstract that is focused only on tumor IDH mutated and the relationship with NAD/NADPH metabolism. In general in different paragraphs of the paper the authors cite tumors IDH mutated. If is this the focus of the manuscript the author have to change the title, non in cancer but in cancer IDH-mutated. I suggest to review also other tumor types and other oncogenic mutations.

The introduction is a repetition of the abstract, the first sentence is not completely real, because tumor cells are more sensitive to NAD inhibition. The problem is to select the tumor type that really addicted to NAMPT inhibition. The author should to first introduce NAMPT, the only real target, and then NAPRT... and not focus attention only on IDH mutated tumor in the introduction.

The paragraph 2 needs to be revised. The first 4 sentences about metabolic reprogramming summerize an old idea of cancer metabolism. Because is now clear that tumor ca adapt their metabolism potentiating glycolysis but also OXPHOS. Moreover, the description of the biosynthesis of NAD is incomplete. Nicotinamide riboside? Add information about this NAD precursor, also in the cartoon.

Could be informative to add a section of the biology/functions of NAD-dependent enzyme, particuparly focusing on NAMPT before paragraph 2.1

Paragraph 3: the sub-paragraphs 3.1 and 3.2 are called "therapeutic role of NAMPT in cancer", change plaese.

I suggest a complete revision of the paper.

Round 2

Reviewer 2 Report

I'm satisfied about the revised version of the manuscript. I think that now is more complete.

I suggest to the authors only to add some references:

NAD-consuming enzymes: PMID:31402913

NAD-NAMPT in cancer and drug resistance: PMID:32010616; PMID:31059816; PMID:30974124; PMID:29330025;PMID:29309612;PMID:29567766